# Intermediate Layers Encode Optimal Biological Representations in Single-Cell Foundation Models

**Vincenzo Yuto Civale**
University of Siena
University of di Firenze
v.civale@student.unisi.it

**Roberto Semeraro**
University of Firenze
roberto.semeraro@unifi.it

**Andrew David Bagdanov**
University of Firenze
andrew.bagdanov@unifi.it

**Alberto Magi**
University of Firenze
alberto.magi@unifi.it

## Abstract

Current single-cell foundation model benchmarks universally extract final layer embeddings, assuming these represent optimal feature spaces. We systematically evaluate layer-wise representations from scFoundation (100M parameters) and Tahoe-X1 (1.3B parameters) across trajectory inference and perturbation response prediction. Our analysis reveals that optimal layers are task-dependent (trajectory peaks at 60% depth, 31% above final layers) and context-dependent (perturbation optima shift 0–96% across T cell activation states). Notably, first-layer embeddings outperform all deeper layers in quiescent cells, challenging assumptions about hierarchical feature abstraction. These findings demonstrate that "where" to extract features matters as much as "what" the model learns, necessitating systematic layer evaluation tailored to biological task and cellular context rather than defaulting to final-layer embeddings. Code to reproduce our results is available at https://github.com/vincenzocivale/scfm-mid-layer-analysis.

## 1 Introduction

Single-cell foundation models (Hao et al., 2024; Cui et al., 2024) pretrained on millions of cells have achieved strong performance across downstream tasks. Two canonical applications are trajectory inference, which recovers continuous biological processes by ordering cells along latent temporal axes, and perturbation response prediction, which evaluates whether representations capture transcriptional effects of genetic or chemical interventions.

Despite these differences, current benchmarks (Kedzierska et al., 2025; Baek et al., 2025; Wang et al., 2025; Atti & Subramaniam, 2025; Csendes et al., 2025; Boiarsky et al., 2024) extract features exclusively from final-layer embeddings, implicitly assuming they provide optimal representations across tasks and biological contexts. However, recent work in NLP and vision shows that intermediate layers often outperform final layers via task-relevant information bottlenecks (Skean et al., 2025), motivating a re-examination of this assumption in biological models.

We systematically evaluate layer-wise embeddings from scFoundation (100M parameters, 12 layers) and Tahoe-X1 (1.3B parameters, 24 layers) across trajectory inference and perturbation response prediction. We find that (i) intermediate layers outperform final embeddings by up to 31% in trajectory inference (stable 60% depth optimum); (ii) perturbation modeling is strongly context-dependent, with optimal layers shifting by up to 96 percentage points across T cell activation states; and (iii) first-layer embeddings can surpass deeper layers in specific contexts. These results challenge the universal reliance on final-layer embeddings and demonstrate that layer selection must be tailored to task and cellular context.

## 2 RELATED WORK

Recent single-cell foundation models such as scFoundation (Hao et al., 2024) and scGPT (Cui et al., 2024) have demonstrated strong performance across downstream tasks including cell type annotation and perturbation prediction. However, existing benchmarks and comparative studies (Kedzierska et al., 2025; Baek et al., 2025; Wang et al., 2025; Atti & Subramaniam, 2025; Csendes et al., 2025; Boiarsky et al., 2024) uniformly extract representations from final network layers, implicitly assuming these provide optimal features across tasks and biological contexts.

In contrast, recent work in natural language processing and vision has shown that intermediate layers often yield superior representations due to task-relevant information bottlenecks (Skean et al., 2025). Despite these findings, layer-wise representation analysis has not been systematically explored in biological foundation models. Our work addresses this gap by evaluating layer-wise embeddings in single-cell models across biologically distinct tasks.

## 3 METHODS

We describe the evaluated foundation models, the datasets and biological tasks used for assessment, and the metrics employed to quantify layer-wise representation quality.

### 3.1 MODELS AND EMBEDDING EXTRACTION

We evaluated two pretrained single-cell foundation models with different scales and architectures. scFoundation (100M parameters) employs an asymmetric encoder-decoder with Read-Depth-Aware pretraining on 50M human cells, processing only expressed genes through vanilla transformers while using Performer blocks for full-sequence reconstruction. Tahoe-X1 (1.3B parameters) uses a scGPT-style transformer encoder pretrained on 266M profiles with masked expression prediction and dual gene/cell-aware decoders (Gandhi et al., 2025).

For each cell, we performed a forward pass through frozen models following standard preprocessing pipelines and extracted hidden states at the output of each transformer block (post-feedforward, after residual addition and layer normalization). To enable cross-model comparison, we normalized layer indices by total depth.

### 3.2 EVALUATION TASKS AND DATASETS

**Trajectory Prediction.** We used the LARRY-based human cord blood LT-scSeq dataset generated in the study by Gao et al. (2025), providing clonally-resolved scRNA-seq of human hematopoietic differentiation with ground-truth temporal annotations. We evaluated whether embeddings preserve pseudotemporal ordering using precomputed diffusion pseudotime (DPT) as reference.

**Perturbation Response.** We used a genome-scale CRISPRi perturb-seq dataset of primary human CD4+ T cells (Zhu et al., 2025), comprising 22M cells with systematic knockdown of 12,748 genes. For computational tractability, we selected donor D1 across three activation states (Rest, Stim8hr, Stim48hr), retaining the 100 most well-represented perturbations per condition by cell coverage. This yielded three evaluation sets spanning transcription factors, signaling molecules, and metabolic regulators, enabling assessment of context-dependent regulatory representation.

See Appendix A for additional dataset details.

#### 3.2.1 EVALUATION METRICS

**Trajectory Preservation.** For each layer $\ell$, we extracted embeddings $\mathbf{E}^{(\ell)} \in \mathbb{R}^{n \times d_\ell}$ and constructed a layer-specific $k$-nearest neighbor graph ($k = 15$, cosine similarity). Using this graph, we computed a layer-derived diffusion pseudotime (DPT) (Haghverdi et al., 2016) and measured its agreement with the reference pseudotime via Spearman correlation:

$$\rho^{(\ell)} = \text{corr}_{\text{Spearman}} \left( \text{DPT}_{\text{ref}}, \text{DPT}^{(\ell)} \right). \tag{1}$$

Values approaching 1 indicate faithful preservation of temporal ordering. Spearman correlation is robust to monotonic transformations and does not assume linear relationships.

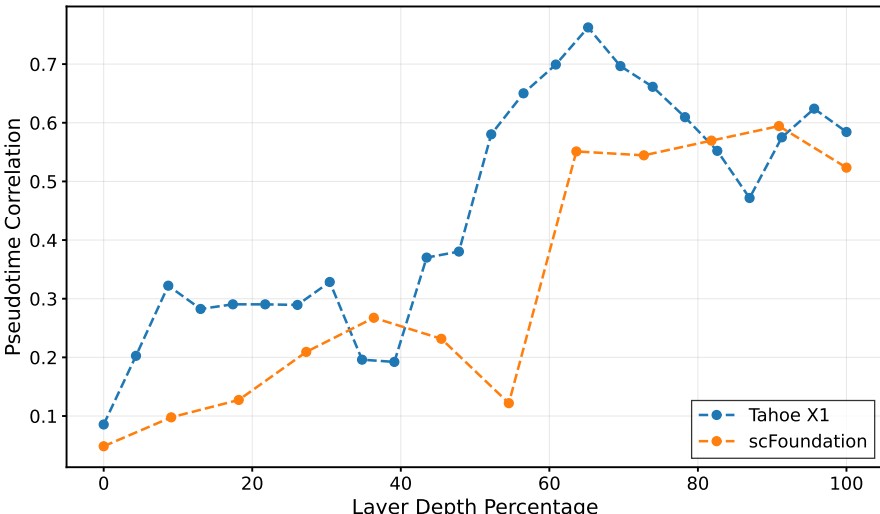

Figure 1: **Layer-wise pseudotime correlation.** Spearman correlation with ground truth pseudotime across normalized depths. Tahoe-X1 (blue) peaks at 60% depth ($\rho = 0.76$, 31% above final layer)

**Perturbation Effect Correlation.** We evaluated layer embeddings using representational similarity analysis. For each perturbation $p$, we computed a differential expression profile $\mathrm{DE}_p$ (log-fold changes via pseudobulk DESeq2 (Love et al., 2014)) and a centroid embedding:

$$\mathbf{c}_p^{(\ell)} = \frac{1}{|C_p|} \sum_{i \in C_p} \mathbf{e}_i^{(\ell)}, \tag{2}$$

where $C_p$ denotes cells with perturbation $p$ and $\mathbf{e}_i^{(\ell)}$ is the layer-$\ell$ embedding of cell $i$. We constructed two similarity matrices: a biological reference $S^{\mathrm{bio}}$ via Spearman correlation of DE profiles, and an embedding-space matrix $S^{(\ell)}$ via cosine similarity of centroids. The layer score measures alignment between these similarity structures:

$$\rho_{\mathrm{pert}}^{(\ell)} = \mathrm{corr}_{\mathrm{Spearman}} \left( \mathrm{vec}(S^{\mathrm{bio}}), \mathrm{vec}(S^{(\ell)}) \right), \tag{3}$$

where $\mathrm{vec}(\cdot)$ extracts upper-triangular entries to avoid redundancy. Higher values indicate that layer embeddings preserve causal regulatory relationships (Kriegeskorte et al., 2008).

## 4 RESULTS

We present a layer-wise analysis of representation quality across two canonical single-cell tasks, highlighting task and context dependent differences in optimal feature depth.

### 4.1 TRAJECTORY PREDICTION REVEALS TASK-DEPENDENT OPTIMAL LAYERS

Figure 1 shows clear layer-wise differences between models. Tahoe-X1 exhibits a pronounced peak at 60% depth (layer 19, $\rho = 0.76$), representing a 31% improvement over its final layer ($\rho = 0.58$) and outperforming scFoundation's best layer by 27%. This intermediate optimum challenges the default reliance on final-layer embeddings.

In contrast, scFoundation displays a largely monotonic increase with depth, peaking at the penultimate layer ($\rho = 0.60$) before a modest drop in the final block. This suggests that even smaller or biologically specialized models may trade off general trajectory structure in their final layers.

Tahoe-X1 shows substantially greater variation across depth (range: 0.08–0.76) than scFoundation (0.05–0.60). Both models perform poorly in shallow layers ($\rho < 0.3$), indicating early blocks

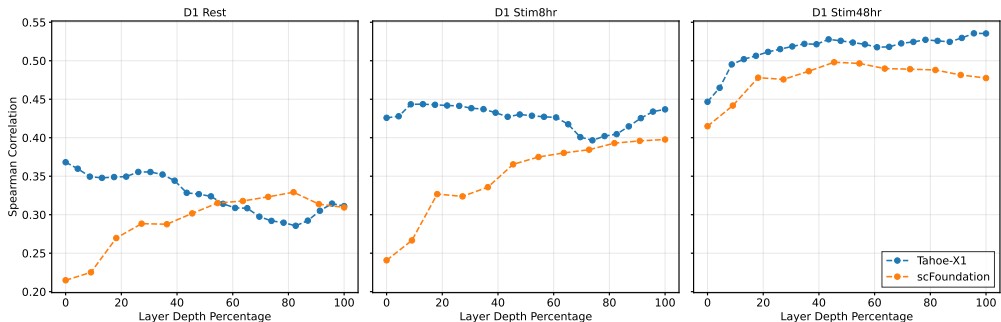

Figure 2: **Layer-wise perturbation response correlation across T cell activation states.** Optimal layers vary dramatically by cellular context, ranging from 0% to 96% depth within the same model. Tahoe-X1 and scFoundation show context-dependent specialization across Rest, Stim8hr, and Stim48hr conditions.

encode low-level or technical signals rather than temporal structure. The sharp increase in Tahoe-X1 between 30–60% depth marks a transition where trajectory-relevant features emerge, consistent with semantic transitions observed in deep language models (Skean et al., 2025).

Beyond the optimum, Tahoe-X1 exhibits a U-shaped profile, with performance degrading at deeper layers ($\rho = 0.47$ at 80% depth) before partial recovery. This pattern suggests late layers increasingly specialize for pretraining objectives at the expense of transferable biological representations, an effect less pronounced in scFoundation. Detailed layer-wise correlation results are provided in Appendix B.

### 4.2 PERTURBATION RESPONSE PREDICTION REVEALS CONTEXT-DEPENDENT LAYER DYNAMICS

Figure 2 reveals dramatic context-dependent dynamics that contrast sharply with trajectory inference. **Optimal depths vary by up to 96 percentage points** within the same model across T cell activation states: Tahoe-X1 peaks at 0% depth in Rest ($\rho = 0.37$), 13% in Stim8hr ($\rho = 0.44$), and 96% in Stim48hr ($\rho = 0.54$). scFoundation exhibits similar variability (82% $\rightarrow$ 100% $\rightarrow$ 45% depth across conditions).

This instability contrasts with trajectory inference, where Tahoe-X1 maintained a stable 60% depth optimum. The first-layer peak in Rest represents the only case across all experiments where shallow representations outperform deeper layers, suggesting perturbation effects in quiescent cells manifest as simple expression shifts not requiring semantic integration.

Performance systematically improves with T cell activation: both models gain ~50% from Rest to Stim48hr (scFoundation: 0.33 $\rightarrow$ 0.50; Tahoe-X1: 0.37 $\rightarrow$ 0.54), suggesting regulatory changes become more coherently represented in activated states. Tahoe-X1's advantage over scFoundation (8–12%) is substantially smaller than in trajectory inference (27%), indicating perturbation modeling may be less sensitive to model scale. Detailed perturbation response results are provided in Appendix C.

## 5 CONCLUSION

We provide the first systematic characterization of layer-wise feature hierarchies in single-cell foundation models, showing that optimal layers depend fundamentally on task and context. Intermediate layers outperform final embeddings by up to 31% in trajectory inference (stable 60% optimum), while perturbation modeling shows strong context dependency (0–96% optimal depth across T cell activation states). These findings challenge benchmarking practices relying solely on final-layer embeddings.

While our analysis covers two models and tasks, broader evaluation across architectures and biological domains is needed. Practitioners should systematically evaluate layers according to task seman-

tics and cellular context. Benchmark frameworks should support all layers, and transfer pipelines should expose layer-specific embeddings. The 31% performance gains highlight that "where" features are extracted matters as much as "what" is learned. As single-cell foundation models scale, understanding layer-wise specialization will be crucial for maximizing biological discovery.

## 6 MEANINGFULNESS STATEMENT

Single-cell foundation models are increasingly used to understand cellular processes, yet all current benchmarks extract features from final layers by default. In this work we show that this assumption fails: optimal representations exist at 60% depth for trajectory inference and shift dramatically (0-96%) across cellular contexts for perturbation modeling. This reveals that "where" we extract features fundamentally determines which biological patterns we can discover. Our findings demonstrate that layer-aware extraction is essential for faithfully representing cellular complexity, directly impacting how researchers should deploy these models to understand development, disease, and therapeutic responses.

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

## APPENDIX A  DATASETS DETAILS

All datasets used in this study are publicly available and were released after 2024, ensuring no overlap with the pretraining data of the evaluated models.

**Trajectory Dataset.** The clonally resolved human hematopoietic differentiation dataset (LT-scSeq, LARRY-based) was obtained from the CLADES study Gao et al. (2025). The raw data is deposited under GEO with accession number GSE276896.

**Perturbation Dataset.** Genome-scale CRISPRi Perturb-seq data of primary human $CD4^+$ T cells were obtained from Zhu et al. (2025). Instructions on how to download the data are available on the official dataset page. In this study, we used donor D1 cells across Rest, Stim8hr, and Stim48hr activation conditions.

**Preprocessing.** All datasets were processed using standard Scanpy pipelines, including quality control filtering, library size normalization, and log-transformation. No dataset-specific fine-tuning or retraining was performed.

## APPENDIX B  TRAJECTORY INFERENCE: LAYER-WISE CORRELATION RESULTS

Tables 1 and 2 present the complete layer-wise Spearman correlations with ground-truth pseudotime for both models. scFoundation achieves its peak performance at layer 11 ($\rho = 0.594$), while Tahoe-X1 peaks at layer 16 ($\rho = 0.763$), corresponding to 60% normalized depth and demonstrating a 31% improvement over its final layer.

Table 1: Layer-wise Spearman correlation ($\rho$) with pseudotime for scFoundation. All correlations are highly significant. (p-value $< 0.001$).

| Transformer Layer | Correlation |
|---|---|
| 1 | 0.0484 |
| 2 | 0.0978 |
| 3 | 0.1274 |
| 4 | 0.2093 |
| 5 | 0.2675 |
| 6 | 0.2316 |
| 7 | 0.1219 |
| 8 | 0.5511 |
| 9 | 0.5445 |
| 10 | 0.5696 |
| **11** | **0.5944** |
| 12 | 0.5235 |

Table 2: Layer-wise Spearman correlation ($\rho$) with pseudotime for Tahoe-X1 (24 layers). All correlations are highly significant. (p-value $< 0.001$).

| Transformer Layer | Correlation |
|---|---|
| 1 | 0.0855 |
| 2 | 0.2026 |
| 3 | 0.3222 |
| 4 | 0.2825 |
| 5 | 0.2904 |
| 6 | 0.2904 |
| 7 | 0.2894 |
| 8 | 0.3285 |
| 9 | 0.1961 |
| 10 | 0.1922 |
| 11 | 0.3701 |
| 12 | 0.3804 |
| 13 | 0.5803 |
| 14 | 0.6502 |
| 15 | 0.6993 |
| **16** | **0.7626** |
| 17 | 0.6966 |
| 18 | 0.6613 |
| 19 | 0.6095 |
| 20 | 0.5521 |
| 21 | 0.4717 |
| 22 | 0.5752 |
| 23 | 0.6240 |
| 24 | 0.5843 |

## APPENDIX C    DETAILED PERTURBATION RESPONSE RESULTS

### C.1    PERTURBATION RESPONSE: SCFOUNDATION

Table 3 shows the layer-wise perturbation effect correlations for scFoundation across three T cell activation states. Optimal layers vary dramatically by cellular context: layer 10 for Rest ($\rho = 0.329$), layer 12 for Stim8hr ($\rho = 0.398$), and layer 6 for Stim48hr ($\rho = 0.498$), corresponding to normalized depths of 82%, 100%, and 45% respectively.

Table 3: **Layer-wise perturbation effect correlation for scFoundation across CD4+ T cell activation states.** Spearman correlation ($\rho$) between embedding-space perturbation effects and ground-truth differential expression signatures. All correlations were highly statistically significant, with p-values far below conventional significance thresholds.

| Transformer Layer | D1_Rest | D1_Stim8hr | D1_Stim48hr |
|---|---|---|---|
| 1 | 0.215 | 0.241 | 0.415 |
| 2 | 0.225 | 0.267 | 0.442 |
| 3 | 0.270 | 0.327 | 0.478 |
| 4 | 0.288 | 0.324 | 0.476 |
| 5 | 0.288 | 0.336 | 0.486 |
| 6 | 0.302 | 0.365 | **0.498** |
| 7 | 0.315 | 0.375 | 0.496 |
| 8 | 0.318 | 0.380 | 0.490 |
| 9 | 0.323 | 0.384 | 0.489 |
| 10 | **0.329** | 0.393 | 0.488 |
| 11 | 0.314 | 0.396 | 0.482 |
| 12 | 0.309 | **0.398** | 0.478 |

### C.2    PERTURBATION RESPONSE: TAHOE-X1

Table 4 presents the layer-wise perturbation effect correlations for Tahoe-X1. The optimal layers span the full model depth: layer 1 for Rest ($\rho = 0.368$, 0% depth), layer 4 for Stim8hr ($\rho = 0.444$, 13% depth), and layer 23 for Stim48hr ($\rho = 0.536$, 96% depth). This 96 percentage point shift in optimal depth across cellular contexts represents the most dramatic context-dependent behavior observed in our study.

Table 4: **Layer-wise perturbation effect correlation for Tahoe-X1 across CD4+ T cell activation states.** Spearman correlation ($\rho$) between embedding-space perturbation effects and ground-truth differential expression signatures. All correlations were highly statistically significant, with p-values far below conventional significance thresholds.

| Transformer Layer | D1_Rest | D1_Stim8hr | D1_Stim48hr |
|---|---|---|---|
| 1 | **0.368** | 0.426 | 0.447 |
| 2 | 0.360 | 0.428 | 0.465 |
| 3 | 0.349 | 0.443 | 0.495 |
| 4 | 0.348 | **0.444** | 0.502 |
| 5 | 0.349 | 0.443 | 0.506 |
| 6 | 0.349 | 0.442 | 0.512 |
| 7 | 0.356 | 0.441 | 0.515 |
| 8 | 0.355 | 0.438 | 0.519 |
| 9 | 0.352 | 0.437 | 0.522 |
| 10 | 0.344 | 0.433 | 0.521 |
| 11 | 0.328 | 0.427 | 0.528 |
| 12 | 0.327 | 0.430 | 0.526 |
| 13 | 0.324 | 0.429 | 0.524 |
| 14 | 0.314 | 0.427 | 0.521 |
| 15 | 0.309 | 0.426 | 0.518 |
| 16 | 0.308 | 0.417 | 0.518 |
| 17 | 0.297 | 0.401 | 0.523 |
| 18 | 0.292 | 0.397 | 0.525 |
| 19 | 0.290 | 0.402 | 0.527 |
| 20 | 0.286 | 0.405 | 0.526 |
| 21 | 0.292 | 0.415 | 0.525 |
| 22 | 0.305 | 0.425 | 0.530 |
| 23 | 0.314 | 0.434 | **0.536** |
| 24 | 0.311 | 0.437 | 0.535 |

