# OpenReview forum: "Intermediate Layers Encode Optimal Biological Representations in Single-Cell Foundation Models"
_ICLR.cc/2026/Workshop/LMRL — ICLR 2026 Workshop LMRL Poster_

### Official Review · Reviewer_1NLX · 2026-02-13
**limited technical analysis report of pre-trained SCFM models**

**Rating:** 3
**Confidence:** 3

**Review:**

The paper questions a common default in single-cell foundation models--using the final-layer embedding as “the” representation. It runs a  layer-by-layer evaluation of embeddings from two pretrained models, scFoundation and Tahoe-X1, and tests which layer is best for two downstream tasks.

The paper provides clear motivation and implementation of the suggested analysis however practical results and justification of claims is very limited, to non existent. First the evaluation is done only on two SCFM models even though it is straightforward and could have easily been extended to additional models. Next even for the given models results are not so groundbreaking and reported with many inconsistencies.  For example, in the trajectory prediction task the big finding is regarding Tahoe-X1, and is being inaccurately reported in multiple places (60% of 24 != layer 16; appendix table != layer 19; main text).
To conclude, while the paper provides an original idea and raises a critical aspect that shall be evaluated in model based biological applications, its significance is very limited given the low n-sample points which do not provide any striking insight.

---

### Official Review · Reviewer_g7ds · 2026-02-25
**Sharp and actionable layer-selection diagnostic for single-cell FMs, with limited scope and proxy-based evidence.**

**Rating:** 7
**Confidence:** 4

**Review:**

The paper systematically evaluates layer-wise representation quality in two single-cell foundation models (scFoundation and Tahoe-X1), instead of using the common default of extracting only the final-layer embedding. It studies two biologically meaningful downstream tasks: trajectory inference and perturbation-response representation quality.

What it proposes/finds:

1. Intermediate layers can substantially outperform final layers for trajectory inference (notably in Tahoe-X1, with a peak around ~60% depth and a large improvement over the final layer).

2. Optimal depth for perturbation-response representation is strongly context-dependent, shifting dramatically across T-cell activation states (Rest / Stim8hr / Stim48hr).

3. In at least one condition (resting/quiescent cells), very shallow layers can outperform deeper layers.

Main claims:

1. Intermediate layers can outperform final layers on trajectory inference.

2. Optimal layer depth depends on task and biological context. Therefore, systematic layer evaluation should replace final-layer defaulting in benchmarking/practice.

3. Layer depth should be considered part of the representation-selection design space in biological FM evaluation.

Overall this is a small, high-value workshop paper with a clear, directly demonstrated diagnostic insight that is immediately useful to the single-cell FM community, but still limited in scope and somewhat over-interpreted in places.

Quality (Technical Soundness):

This is a technically coherent diagnostic study with an appropriate workshop scope. The paper’s strongest point is the directness of the core evidence: the claimed layer-wise effects are visible in the reported layer curves/tables and are large enough to matter in practice.

What is shown:

1. The authors implement layer-wise extraction across two real pretrained models and compare performance across all layers.

2. The trajectory task uses a pseudotime-preservation metric (Spearman correlation with DPT-derived structure), which is a reasonable task-aligned proxy for trajectory quality.

3. The perturbation task uses an RSA-style correlation between biological perturbation similarity (derived from DE/pseudobulk effects) and embedding similarity, which is a sensible representational alignment metric for this setting.

4. The headline observations (trajectory intermediate peak; perturbation context-dependent shifts; shallow-layer success in resting cells) are directly supported by the layer-wise empirical results.

What is inferred (reasonable, but indirect):

1. Interpretations such as “late layers specialize to the pretraining objective” and “resting cells favor shallow layers because effects are simpler” are plausible, but currently hypotheses rather than demonstrated mechanisms.

2. Claims about hierarchical abstraction across layers are suggestive but under-tested given only two models and limited task diversity.

What is overstated / should be calibrated:

1. The perturbation RSA metric measures representational alignment with perturbation-derived similarity structure, not recovery of causal regulatory relationships. “Causal” language should be removed or softened.

2. Broad statements implying the paper “fundamentally determines” what biological patterns can be discovered are too strong for the current evidence base (two models / two tasks / one donor in perturbation analysis).

3. “First systematic characterization” should be qualified (the contribution is best framed as a systematic single-cell FM study, not the first layer-probing study in biology/ML broadly).

Uncertainty / robustness / statistical issues:

1. A major weakness is the lack of strong uncertainty quantification for layer-to-layer differences (e.g., bootstrap CIs or resampling-based uncertainty bands on the layer curves). Reporting correlation significance alone is not enough to establish that the differences between layers are robust.

2. Perturbation analysis is restricted to one donor (D1), so donor-specific effects cannot be ruled out.

3. Sensitivity to design choices (e.g., DPT settings/root choice, kNN graph parameters, perturbation subset size, similarity metric choice) is not deeply explored.

Overall, the paper is sound as a diagnostic note, but it should be interpreted as a compelling call-to-action rather than a definitive characterization.

Clarity:

1. The paper is clearly written and easy to follow, especially given its short length.

2. The motivation (why final-layer defaulting is a hidden assumption) is well articulated.

3. Figures and layer-wise trends are easy to interpret, and the appendix tables substantially improve transparency and reproducibility.

4. Metric definitions are reasonably clear, though some readers may want more detail on the perturbation RSA construction and sensitivity choices.

What remains underdeveloped:

1. Why the observed layer shifts occur biologically/model-wise.

2. How practitioners should choose layers for new tasks beyond “evaluate all layers.”

3. The distinction between evidence-backed observations and speculative interpretation.

Originality: Its originality lies in a diagnostic reframing- layer selection is itself a biologically consequential design choice. Final-layer defaults can produce misleading conclusions, and optimal representation depth can depend on biological state/context. This is a meaningful contribution to LMRL, especially in single-cell foundation model practice.

Significance: Moderate to high. The reported gains/shifts are large enough to change evaluation and downstream pipeline practice.
The recommendation to evaluate multiple layers (rather than defaulting to final-layer embeddings) is immediately actionable and low-cost.
The paper clearly demonstrates a phenomenon but does not yet explain the underlying representational/biological mechanisms. It opens an important line of inquiry about depth-specific biological information encoding.

LMRL relevance: Strong. The paper directly addresses where “meaningful” biological representations reside in a model and challenges the assumption that meaningfulness is concentrated in the final layer.

Pros:

1. Clear, focused diagnostic contribution with strong practical relevance.

2. Direct empirical challenge to a widespread benchmarking default (final-layer extraction).

3. Large and memorable effects (trajectory improvement and context-dependent depth shifts).

4. Biologically grounded tasks (trajectory and perturbation response) rather than arbitrary probes.

5. Layer-wise appendix tables improve transparency and reproducibility.

Good workshop fit: useful negative result / benchmarking insight without requiring a new architecture.

Cons:

1. Evidence scope is narrow (two models, limited tasks, one donor in perturbation analysis).

2. Limited uncertainty quantification for layer-wise differences and optimal-layer estimates.

3. Some interpretations are speculative and not directly tested.

4. “Causal/mechanistic” wording is too strong for the perturbation metric.

5. Little sensitivity analysis for evaluation design choices (DPT setup, perturbation subset, similarity metrics).

6. No deeper probing of what is encoded at different layers or why the layer transitions occur.

Reccomendation rationale: The strongest contribution is a directly shown diagnostic insight: optimal biological representation depth in single-cell foundation models is task- and context-dependent, and final-layer defaults can be materially suboptimal. This is exactly the kind of actionable, biologically grounded benchmark insight workshops should reward. I am not rating it higher because the evidence base is still narrow, uncertainty quantification is limited, and some interpretive/causal language is stronger than what the metrics support.

---

### Official Review · Reviewer_bMuU · 2026-02-25
**Review for Intermediate Layers Encode Optimal Biological Representations in Single-Cell Foundation Models**

**Rating:** 5
**Confidence:** 5

**Review:**

The authors evaluate the performance of single cell foundation models (scFMs), scFoundation and Tahoe X1, on two tasks, trajectory inference and perturbation prediction, to determine what layers are optimal for embedding. They find that different layers for each model have significantly better performance than the final layer, which is important because models by default use the final layer to produce embeddings.

While it is relevant to model users to know this, this is generally well known, and not evaluated rigorously enough in this paper, given only two datasets, and two models are measured. The paper does not offer any actionable insights.

**Notes:**

The related work section is very sparse and simply repeats the introduction including seemingly the exact same citations.

It's unclear exactly how the embeddings for each layer are being generated. Is the embedding the mean across every token in the layer, or the mean of the CLS token?

---

### Meta-Review · Area_Chair_fekA · 2026-02-25

**Recommendation:** Accept (Poster)
**Confidence:** 4

**Metareview:**

Accept

---

### Decision · Program_Chairs · 2026-03-02

**Decision:**

Accept (Poster)

**Comment:**

Please see the meta-review.